# Patients with Initial Negative RT-PCR and Typical Imaging of COVID-19: Clinical Implications

**DOI:** 10.3390/jcm9093014

**Published:** 2020-09-18

**Authors:** Florent Baicry, Pierrick Le Borgne, Thibaut Fabacher, Martin Behr, Elena Laura Lemaitre, Paul-Albert Gayol, Sébastien Harscoat, Nirvan Issur, Sabrina Garnier-Kepka, Mickael Ohana, Pascal Bilbault, Mathieu Oberlin

**Affiliations:** 1Emergency Department, Hôpitaux Universitaires de Strasbourg, 67000 Strasbourg, France; florent.baicry@chru-strasbourg.fr (F.B.); pierrick.leborgne@chru-strasbourg.fr (P.L.B.); martin.behr@chru-strasbourg.fr (M.B.); elenalaura.lemaitre@chru-strasbourg.fr (E.L.L.); paul-albert.gayol@chru-strasbourg.fr (P.-A.G.); sebastien.harscoat@chru-strasbourg.fr (S.H.); nirvan.issur@chru-strasbourg.fr (N.I.); sabrina.garnier-kepka@chru-strasbourg.fr (S.G.-K.); pascal.bilbault@chru-strasbourg.fr (P.B.); 2INSERM (French National Institute of Health and Medical Research), UMR 1260, Regenerative NanoMedicine (RNM), Fédération de Médecine Translationnelle (FMTS), University of Strasbourg, 67000 Strasbourg, France; 3ICube, équipe IMAGeS, UMR7357, Université de Strasbourg, 67000 Strasbourg, France; thibaut.fabacher@chru-strasbourg.fr; 4Radiology Department, Nouvel Hôpital Civil, Strasbourg University Hospital, 67000 Strasbourg, France; mickael.ohana@chru-strasbourg.fr; 5HuManiS Laboratory (EA7308), Ecole de Management (EM), University of Strasbourg, 67000 Strasbourg, France

**Keywords:** COVID-19, reverse transcriptase polymerase chain reaction, clinical management, Bayesian analysis

## Abstract

The sensitivity of reverse transcriptase polymerase chain reaction (RT-PCR) has been questioned due to negative results in some patients who were strongly suspected of having coronavirus disease 2019 (COVID-19). The aim of our study was to analyze the prognosis of infected patients with initial negative RT-PCR in the emergency department (ED) during the COVID-19 outbreak. This study included two cohorts of adult inpatients admitted into the ED. All patients who were suspected to be infected with SARS-CoV-2 and who underwent a typical chest CT imaging were included. Thus, we studied two distinct cohorts: patients with positive RT-PCR (PCR+) and those with negative initial RT-PCR (PCR–). The data were analyzed using Bayesian methods. We included 66 patients in the PCR– group and 198 in the PCR+ group. The baseline characteristics did not differ except in terms of a proportion of lower chronic respiratory disease in the PCR– group. We noted a less severe clinical presentation in the PCR– group (lower respiratory rate, lower oxygen need and mechanical ventilation requirement). Hospital mortality (9.1% vs. 9.6%) did not differ between the two groups. Despite an initially less serious clinical presentation, the mortality of patients infected by SARS-CoV-2 with a negative RT-PCR did not differ from those with positive RT-PCR.

## 1. Introduction

The 2019 novel coronavirus causing Severe Acute Respiratory Syndrome coronavirus 2 (SARS-CoV-2) was reported for the first time in December 2019 in Wuhan, China [1]. The outbreak rapidly evolved into a global pandemic resulting in worldwide concern. Due to the increasing number of coronavirus disease 2019 (COVID-19) patients, rapid and accurate detection of the virus is important [2]. Detection of viral nucleic acids by reverse transcriptase polymerase chain reaction (RT-PCR) is considered the gold standard for the diagnosis of SARS-CoV-2 [3]. However, the sensitivity of RT-PCR has been questioned due to negative results in patients with a strong suspicion of having the disease based on clinical presentation, exposure history and chest computed tomography (CT) imaging in the epidemic phase [4,5,6,7]. Due to viral load variability, a negative RT-PCR result does not completely rule out the presence of COVID-19 (sensitivity between 83% and 93%) [7]. Negative results could result from non-recommended sampling techniques, laboratory error during sampling or viral genome mutations [2,8,9]. Several studies have reported that RT-PCR may become positive after initially negative nasopharyngeal swabs [4,10]. In addition to the technical problem and genetic diversity of this new coronavirus, viral load kinetics at different anatomical sites in patients could contribute to false-negative results [8,11]. Viral load has been described as high early in the course of the virus and peaked around day six in throat and sputum specimens [7,12]. The viral load appears to be higher in bronchoalveolar fluid washings than in sputum, throat swabs and nasal passages [11,12].

The management of patients with typical clinical signs of SARS-CoV-2 pneumonia and typical chest CT imaging without RT-PCR results or with negative RT-PCR raises some pragmatic questions about, for example, isolation, orientation and prognosis [13]. Clinical symptoms between patients with positive and negative RT-PCR results appear to be similar, but few studies have compared progression and prognosis [14]. Some authors have suggested that positive results after an initial negative swab test are consistent with disease progression [12]. The aim of our study was to analyze the prognosis and evolution of patients with COVID-19 with initial negative RT-PCR during the outbreak.

## 2. Material and Methods

We used the prospective cohort called COVIDHUS, which includes all suspect patients of COVID-19 admitted to a university hospital during the COVID-19 epidemic. Our region was one of the most impacted by the pandemic in Europe, with nearly 3400 deaths and more than 10,000 patients infected at the end of May. This study included two cohorts of adult inpatients (>18 years old) admitted to the emergency department (ED) of a university hospital. We screened all the patients suspected of being infected with SARS-CoV-2 between 14 March and 16 April 2020 and selected those with a typical chest CT imaging. An RT-PCR test on nasal and pharyngeal swabs was performed in the emergency department, allowing us to identify two distinct patient cohorts: those with positive initial RT-PCR (PCR+ group) and those with negative initial RT-PCR (PCR− group). We selected patients in the PCR+ group by matching the patients by gender and age. The PCR−/PCR+ group ratio was one to three. 

In our study, COVID-19 was defined by the presence of the following three criteria: (a) epidemiological history (history of travel to and/or residence in an area with a high prevalence of COVID-19 or history of exposure to patients with COVID-19 and symptoms within 14 days prior to onset of illness); (b) clinical manifestations (fever, myalgia, cough, dyspnea and/or normal or decreased white blood cell count or decreased lymphocyte count); (c) chest CT with typical lesions (ground-glass opacities and/or alveolar consolidation, with bilateral involvement and peripheral distribution) [15]. According to Pascarella et al. [16], SARS-CoV-2 infection was defined during the outbreak as a positive result of RT-PCR test of nasal and pharyngeal swabs or a chest CT interpreted as typical by a radiologist. 

In the ED, a chest CT scan and a nasopharyngeal swab were performed immediately at arrival in all patients admitted for COVID-19 clinical suspicion. The chest CT performed in the ED was interpreted by an experienced radiologist and classified as typical, indeterminate or incompatible with COVID-19. According to the European Society of Radiology and the European Society of Thoracic Imaging, the severity of lung damage was classified as minimal (<10%), moderate (10–24%), extensive (25–49%) and severe (≥50%) [17]. Patients with chest CT scans interpreted as normal, incompatible or indeterminate were excluded. Only patients with typical lesions on chest CT were included. 

The nasopharyngeal swab was collected by a trained team (ED head physician) according to a detailed video in a previously published study [18]. Quantitative real-time reverse transcriptase PCR (qRT-PCR) tests for SARS-CoV-2 nucleic acid were performed on nasopharyngeal swabs. Primer and probe sequences target two regions of the RdRp gene and are specific to SARS-CoV-2. The sensitivity of the assay is approximately 10 copies per reaction [19].

Data were collected from electronic medical records using a standardized consent report form. These data included epidemiological, clinical, laboratory and radiological data. The date of the onset of the symptoms (fever, myalgia, cough, dyspnea, etc.) before the visit to the ED was collected. Clinical parameters were also recorded during the visit to the ED. Minimum oxygen saturation, blood pressure, heart rate, respiratory rate, oxygen requirement and Glasgow coma scale were recorded. The main medical histories that may affect prognosis were collected: pre-existing respiratory, cardiovascular or kidney diseases, active cancer, diabetes, hypertension and obesity [16,20,21,22]. A routine blood examination was performed, including a complete blood count, and serum biochemical testing (renal and liver function, electrolytes, serum creatinine, C-reactive protein). The Sepsis-related Organ FAilure (SOFA) score was calculated according to the method described by Vincent et al. [23]. The intake service was scored. When patients were admitted or transferred to the Intensive Care Unit (ICU), the need for mechanical ventilation was noted. In-hospital mortality and date of discharge from the hospital were reported. The study was approved by the local ethics committee (University of Strasbourg, n°CE-2020-43), and the data processing was notified to the French data protection authority (CNIL, n°2217715). 

## 3. Statistical Analysis

The data were analyzed using Bayesian methods. The data were described as frequency (%) for categorical data and with mean and standard deviation (SD) for quantitative data. To compare the two groups, we used mean comparison with Gaussian prior distribution for quantitative data and proportion comparison with Beta prior distribution for categorical data. Priors were defined before the study and based on mild assumptions derived from expert knowledge and available literature. Results are provided as point estimates of means or proportions difference with their associated 95% credibility interval. We computed the probability that the between-group difference is larger than 0 (Pr(diff > 0)). We remind the reader that Bayesian methods do not use *p*-values and that the computed probabilities should not be confused with *p*-values. Probabilities near 1 or 0 suggest an effect, respectively, of a positive or negative difference. All computations were done with R 4.0 and JAGS software(s) with all the required additional packages.

## 4. Results

Between 14 March and 16 April 2020, a total of 2329 patients visited our ED, of whom 1671 were suspected of having COVID-19. Among them, 680 patients had chest CT imaging typical of COVID-19. Of these, 66 had an initial negative RT-PCR and 447 a positive RT-PCR. We included all 66 patients with negative RT-PCR (PCR− group) and randomly matched them with 198 patients with initial positive RT-PCR (PCR+ group). In the PCR– group, 32 patients became positive during the length of their hospital stay (LOS) and 34 patients remained negative during their whole stay (Figure 1). The mean number of RT-PCR was 2.1 (SD: 1.7) in both groups. All patients were considered and treated for COVID-19 during their hospital stay. The mean duration of symptoms from the onset to the ED visit was similar at 8.8 days (SD: 6.6) for the PCR– group and 8.2 days (SD: 6.1) for the PCR+ group (Pr(Diff > 0) = 0.75).

The baseline characteristics are presented in Table 1. The mean age was 59.7 (SD: 15.3) in the PCR− group and 59.9 years old (SD 15.9) in the PCR− group (Pr(diff > 0 = 0.47), ranging from 18 to 100 years old. Men represented 62.1% of patients in the PCR− group and 64.6% in the PCR+ group (Pr(Diff > 0) = 0.35). The mean BMI was lower in the PCR− group (25.0 Kg/m^2^ vs. 27.5 Kg/m^2^ (Pr(diff > 0 = 0.09). The most common comorbidities were hypertension (43.9% in the PCR– group vs. 43.3% in the PCR+ group, Pr(diff>0) = 0.53), followed by diabetes (21.2% vs. 23.2%, Pr(Diff > 0) = 0.39) and active cancer or hemopathy (16.7% vs. 12.6%, Pr(diff > 0) = 0.81)) without any significant difference. The number of patients with chronic respiratory disease was higher in the case group (PCR–) than in the PCR+ group (30.3% vs. 11.6%, Pr(diff > 0) > 0.99). 

The mean respiratory rate and the mean oxygen need were lower in the PCR− group (23.8 breaths per minute (SD 5.7) vs. 27.1 breaths per minute (SD 9.1), Pr(Diff > 0) < 0.01 and 3.0 L/min vs. 4.4 L/min, Pr(diff> 0) < 0.01 respectively). Conversely, the proportion of patients with heart rate ≥120 bpm was higher in the PCR− group than in the PCR+ group (12.1% vs. 6.6%, Pr(diff > 0) = 0.93). The biological and imaging characteristics are presented in Table 2. Concerning laboratory findings, we did not find any statistically and clinically significant difference between the two groups, especially in the inflammatory or renal parameters. Regarding the imaging features, we noted a trend of a lower proportion of patients with extensive or severe COVID-19 patterns in the PCR– group (12.5% and 14.1%, respectively) than in the PCR− group (21.1%, Pr(diff > 0) = 0.06 and 20.6%, Pr(diff > 0) = 0.13 respectively). In contrast, we observed a higher proportion of mild COVID-19 patterns in the PCR+ group (26.6% vs. 13.4%, Pr(Diff > 0) = 0.99). The SOFA score was lower on the PCR− group (5.0 (SD 1.7) vs. 5.6 (SD 1.7), Pr(Diff > 0) = 0.01). The number of patients requiring mechanical ventilation was also lower in the PCR− group (7.6% vs. 14.7%, Pr(Diff > 0) = 0.08).

Nevertheless, no difference was found in the in-hospital mortality rate, with 9.1% (CrI = (4.3;18.6)) in the PCR– group vs. 9.6% (CrI = (6.2;14.5)) in the PCR+ group (Pr(Diff > 0) = 0.50). Similarly, the LOS was not significantly different between the two groups (8.7 days (SD 7.4) in the PCR– group, vs. 9.8 days (SD 8.3) in the PCR+ group, (Pr(diff > 0) = 0.17). On the other hand, we noted that the proportion of patients requiring ICU admission tended to be lower in the PCR– group (22.7% vs. 30.8%, (Pr(diff > 0) = 0.11) (Table 3).

## 5. Discussion

With more than 70% of patients consulting for suspected SARS-CoV-2 pneumonia during the COVID-19 epidemy, our hospital was hardly impacted. The study population was mostly a symptomatic hospitalized population. This study answers a pragmatic question in this specific population: Do patients with SARS-CoV-2 pneumonia and an initial negative RT-PCR test have a different outcome?

During the epidemic phase, the diagnosis of COVID-19 was made by chest CT scan with a higher sensitivity than RT-PCR. In a recent meta-analysis, Kim et al. [7] showed that diagnostic performance of CT-scan depends on the prevalence of the disease. Moreover, in areas with high prevalence, pragmatic protocols using chest CT to help diagnose COVID-19 have been implemented to facilitate the triage of patients in the ED [13]. Thus, in our organization during the epidemic phase, we considered patients with compatible clinical characteristics and typical imaging as infected by SARS-CoV-2 even with an initial negative RT-PCR [24]. While the relevance of chest as a screening tool has been largely criticized [25,26,27], a recent national registry [28] demonstrated a high diagnostic performance of chest CT and advocates the use of this examination as a first-line screening tool in clinical COVID-19 suspicion in the pandemic context.

There can be many reasons for a negative RT-PCR (fluctuating virus load, non-recommended sampling techniques, mutations in the viral genome) [2,8,9]. In our study, nasopharyngeal swabs were performed by the same trained team throughout the study, which reduced the risk of unrecommended sampling techniques. Moreover, the duration of symptoms from onset to the ED was the same in both groups. Thus, the viral load kinetics do not seem to be a valid explanation. Furthermore, the proportion of false-negative RT-PCR was estimated to be more than 21% with a mean symptom duration of 8 days [29].

The mean age was 60 years, with a higher proportion of males (64%), as shown in previous studies [22,30]. The main common comorbidities were hypertension, diabetes, chronic respiratory disease, and active cancer or hemopathy. The characteristics of our population are consistent with those usually found in patients with COVID-19 [21,29]. The patients’ comorbidity profiles were similar in both groups except for a lower proportion of chronic respiratory diseases in the PCR– group. The overall in-hospital mortality rate was similar in both groups at 9.5%. This mortality rate is consistent with previous studies that showed a mortality rate between 3.2% and 28.3% [20,22,29,30,31,32,33].

During visits to the ED, clinical respiratory parameters were significantly worse in the control group, with a mean respiratory rate of 27.1 vs. 23.8 breaths per minute. Patients in the control group required more oxygen than patients in the PCR– group. For other clinical parameters, no difference was observed. Li et al. recently suggested that patients with negative RT-PCR had the same clinical characteristics as patients with positive RT-PCR [14]. Their study compared clinical and biological characteristics and found only a significant difference in the proportion of patients with dyspnea. It did not include objective respiratory parameters (respiratory rate, oxygen saturation). These data may suggest that initially, the severity of respiratory impairment in the patients with negative RT-PCR was lower than those with positive RT-PCR. Regarding the imaging results of chest CT, we found the same trend with a higher proportion of extensive and severe imaging in the control group. Finally, the SOFA score was higher in the control group. We can hypothesize that patients with a more severe clinical presentation may have a higher viral load at the time of their visit and therefore a higher proportion of positive RT-PCR. The proportion of patients requiring admission to ICU tends to be higher in the PCR+ group. The need for invasive mechanical ventilation was also higher. However, there was no difference in in-hospital mortality between the two groups (9.1% in the PCR− group vs. 9.6% in the PCR+ group) or in the length of stay. The main hypothesis remains the concentration of viral copies excreted by the patients, which would be higher in the PCR+ group. This hypothesis and its underlying pathophysiological causes have yet to be verified by further studies. 

A limitation of our study could be a selection bias in defining COVID-19 with negative RT-PCR and typical chest CT lesions. Although we were in the middle of the pandemic, typical chest CT lesions could be another type of infection, even if it was a low probability. Half of the PCR– group turned positive after several nasal and pharyngeal swabs. The other half were considered as typical COVID-19 for physicians in this context of the high prevalence of the disease. This definition of COVID-19 diagnosis could not be used in case of lower prevalence and must be kept in mind for clinicians.

Although single-centered and on a limited sample, our study answers a pragmatic question that arises in all emergency departments in the epidemic phase. 

## 6. Conclusions

Despite an initially less severe clinical presentation, less need for mechanical ventilation and ICU admission, mortality and length of stay of SARS CoV-2-infected patients with negative RT-PCR did not differ from patients with positive RT-PCR. The management of such patients should be the same in the ED as for those with PCR+.

## Figures and Tables

**Figure 1 jcm-09-03014-f001:**
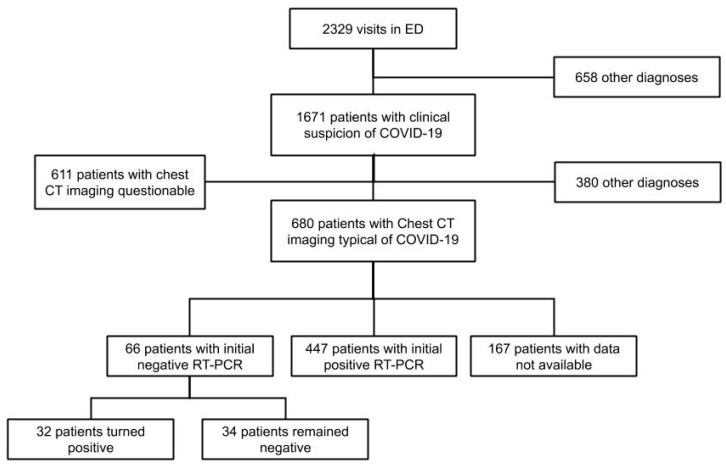
Flow-chart. Abbreviations: ED = emergency department; CT = computed tomography; COVID-19 = Coronavirus Disease 2019, RT-PCR = reverse transcriptase polymerase chain reaction.

**Table 1 jcm-09-03014-t001:** Clinical characteristics.

Parameters	PCR− Group	PCR+ Group	Difference in Means or Proportion	Pr(Diff > 0) ^b^
	No = 66	No = 198	(CrI) ^a^	
**Population**				
Age, years, means (SD)	59.7 (15.3)	59.9 (15.9)	−0.17 (−4.13;3.73)	0.47
Gender, Male	41 (62.1)	128 (64.6)	−0.03 (−0.16;0.10)	0.35
**Comorbidity**				
BMI, kg/m^2^, mean (SD)	25.0 (10.9)	27.5 (10.4)	−2.20 (−5.41;1.00)	0.09 *
Hypertension	29 (43.9)	86 (43.4)	0.01 (−0.13;0.14)	0.53
Diabetes mellitus	14 (21.2)	46 (23.2)	−0.01 (−0.12;0.10)	0.39
Active smoking	7 (10.6)	11 (5.6)	0.06 (−0.02;0.15)	0.93
Chronic heart disease	5 (7.6)	13 (6.6)	0.02 (−0.05;0.10)	0.66
Chronic respiratory disease	20 (30.3)	23 (11.6)	0.19 (0.07;0.31)	>0.99 *
Asthma	7 (10.6)	12 (6.1)	0.05 (−0.02;0.14)	0.90 *
Chronic bronchitis	8 (12.1)	4 (2.0)	0.11 (0.03;0.20)	0.99 *
Chronic kidney disease	4 (6.1)	17 (8.6)	−0.02 (−0.08;0.06)	0.31
Active cancer or hemopathy	11 (16.7)	25 (12.6)	0.05 (−0.05;0.15)	0.81
**Clinical features in the ED**				
Time from onset to admission				
day, mean (SD)	8.8 (6.6)	8.2 (6.1)	0.62 (−1.13;2.34)	0.75
≥7 days	34 (51.5)	87 (43.9)	0.07 (−0.06;0.21)	0.86
Systolic Blood Pressure				
mmHg, mean (SD)	135.9 (24.6)	134.5 (21.3)	1.13 (−4.93;7.19)	0.64
Heart rate				
bpm, mean (SD)	92.9 (19.0)	92.8 (16.5)	0.10 (−4.63;4.90)	0.52
≥120 bpm	8 (12.1)	13 (6.6)	0.06 (−0.02; 0.16)	0.93 *
Glasgow coma scale	15 (0.1)	14.9 (0.4)	0.08 (0.01;0.15)	0.98
Respiratory rate				
breaths per minute, mean (SD)	23.8 (5.7)	27.1 (9.1)	−3.16 (−5.42;−0.87)	<0.01 *
≥24 breaths per minute	21 (50.0)	99 (68.8)	−0.18 (−0.35;0.02)	0.01 *
Minimal oxygen saturation at admission				
%, mean (SD)	92.1 (7.5)	91.4 (8.6)	0.76 (−1.44;2.89)	0.75
<94%	31 (47.0)	96 (48.5)	−0.01 (−0.15;0.12)	0.42
Maximal oxygen level at admission				
L/min, mean (SD)	3.0 (3.5)	4.4 (4.4)	−1.37 (−2.39;−0.34)	<0.01 *
≥5 L/min	10 (15.2)	65 (32.8)	−0.17 (−0.27;−0.05)	<0.01 *

Data are all expressed in mean (SD) or n/No (%) where No is the total number of patients with available data. Abbreviations: SD = standard deviation; BMI = body mass index; Kg/m^2^ = kilograms per square meter; ED = emergency department; mmHg = millimeters of mercury; BPM = beats per minute; L/min = liters per minute; * probability > 90% or < 10% that difference is larger than 0. ^a^ (CrI) = credibility interval ^b^ Pr(diff > 0)= probability that the difference is larger than 0.

**Table 2 jcm-09-03014-t002:** Biological and imaging characteristics.

Parameters	PCR− Group	PCR+ Group	Difference in Means or Proportion	Pr(Diff > 0) ^b^
	No = 66	No = 198	(CrI) ^a^	
**Laboratory Findings**				
Neutrophil count, per µL, mean (SD)	5294.8 (3419.4)	5187.8 (2752.4)	4.71 (−171.35;175.26)	0.53
Lymphocyte count, per µL, mean (SD)	1083.6 (513.5)	1054.8 (629.4)	9.50 (−78.60;96.02)	0.58
Hemoglobin, g/dL, mean (SD)	16.2 (16.7)	13.6 (1.8)	2.38 (−1.38;6.11)	0.89
Platelet count, per µL, mean (SD)	223.0 (99.8)	215.8 (78.9)	4.17 (−15.5;23.62)	0.66
Natremia, mmol/L, mean (SD)	136.3 (3.5)	135.3 (3.5)	0.94 (−0.03;1.91)	0.97 *
Serum creatinine, µmol/L, mean (SD)	81.4 (39.1)	85.8 (57.4)	−3.63 (−15.13;7.85)	0.27
ALT, U/L, mean (SD)	46.4 (33.5)	88.0 (63.4)	−36.26 (−88.03;13.94)	0.08 *
AST, U/L, mean (SD)	50.0 (33.5)	63.3 (60.2)	−11.19 (−23.19;0.92)	0.03 *
**C−Reactive Protein**				
mg/L, mean (SD)	96.7 (81.7)	113.2 (83.9)	−10.10 (−27.98;8.80)	0.14
inf 100 mg/L	39 (61.9)	101 (51)	0.10 (−0.03;0.24)	0.93 *
100−200mg/L	16 (25.4)	70 (35.4)	−0.09 (−0.21;0.04)	0.08 *
≥200 mg/L	8 (12.7)	27 (13.6)	−0.01 (−0.09;0.10)	0.47
Lactate				
mmol/L, mean (SD)	1.3 (1.1)	1.1 (0.6)	0.21 (−0.16;0.57)	0.87
>2 mmol/L	4 (9.5)	10 (7.4)	0.03 (−0.06;0.15)	0.73
**Imaging**				
Mild	17 (26.6)	26 (13.0)	0.14 (0.03;0.27)	0.99 *
Moderate	30 (46.9)	87 (44.8)	0.01 (−0.12;0.15)	0.57
Extended	8 (12.5)	41 (21.1)	−0.08 (−0.17;0.02)	0.06 *
Severe to critical	9 (14.1)	40 (20.6)	−0.06 (−0.16;0.05)	0.13
**Severity**				
SOFA score	5.0 (1.7)	5.6 (1.7)	−0.55 (−1.04;0.08)	0.01 *
Mechanical ventilation	5 (7.6)	29 (14.7)	−0.06 (0.14;0.03)	0.08 *

Data are all expressed in mean (SD) or n/No (%) where No is the total number of patients with available data. Abbreviations: SD = standard deviation; µL = microliter; g/dL = grams per deciliter; mmol/L = millimoles per liter; mg/L = milligrams per liter; ALT = alanine aminotransferase; AST = aspartate aminotransferase; SOFA = sepsis-related organ failure; * probability > 90% or < 10% that difference is larger than 0. ^a^ (CrI) = credibility interval ^b^ Pr(diff > 0) = probability that the difference is larger than 0.

**Table 3 jcm-09-03014-t003:** Outcome.

Outcome	PCR− Group	PCR+ Group	Difference in Means or Proportion	Pr(Diff > 0) ^b^
	No = 66	No = 198	(CrI) ^a^	
In−hospital mortality, n (%)	6 (9.1)	19 (9.6)	0.01 (−0.07;0.09)	0.50
ICU admission, n (%)	15 (22.7)	61 (30.8)	−0.07 (−0.19;0.05)	0.11
Hospital LOS, days, mean (SD)	8.7 (7.4)	9.8 (8.3)	−1.01 (−3.06;1.16)	0.17

^a^ (CrI) = credibility interval; ^b^ Pr(diff > 0) = probability that the difference is larger than 0. Data are all expressed in mean (SD) or n/No (%) where No is the total number of patients with available data. Abbreviations: SD = standard deviation; ICU = intensive care unit; LOS = length of stay.

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
