# Peer review of "Patients with Initial Negative RT-PCR and Typical Imaging of COVID-19: Clinical Implications"

_jcm, 2020, doi:10.3390/jcm9093014_

Round 1
Reviewer 1 Report
In this study a group of Covid-19 patients with clear signs of pneumonia on CT and initial negative RT-PCR test are compared with a matching group of Covid-19 pneumonia patients with positive RT-PCR test.
My questions/remarks:
- Material and Methods p2 lines 67-72: did these criteria need to be present simultaneously or was a positive chest CT enough?
- Material and Methods p2 line 75: was in some patients more than one CT scan performed and if so which one was scored by the radiologist?
- Results p 3 lines 115: when was the initial RT-PCR performed? Was this at the time patient was admitted to the ED and at the same moment that the CT was performed or was there a time difference between RT-PCR test and CT?
- Results P3 line 32: It would be interesting to know when the second test was performed and when the 32 negative patients finally tested positive
- It is not clear to me what the clinical value of the findings in this study is. Both groups had a clear diagnosis of Covid-19 pneumonia and adequate treatment could be started and adapted to the clinical evolution. In addition not so many differences between the two groups were found. It is also well known as stated in the introduction that a RT-PCR test can be negative while CT shows clear signs of Covid-19 pneumonia.
Author Response
Dear reviewer 1
Please find below our point-to-point answers to your questions and remarks. We are very grateful for the time you spend to read and comment our work.
Material and Methods p2 lines 67-72: did these criteria need to be present simultaneously or was a positive chest CT enough?
Exact, we change and precise it in our material and methods p2 lines 67-72: COVID-19 was defined with the presence of the three following criteria: epidemiological history, clinical manifestations and typical parenchymal lesions on chest CT.
Material and Methods p2 line 75: was in some patients more than one CT scan performed and if so which one was scored by the radiologist?
Thank you for your comment. Only the initial chest CT- i.e. the one acquired at the ED, was used. We precise this point on line 76-77
Results p 3 lines 115: when was the initial RT-PCR performed? Was this at the time patient was admitted to the ED and at the same moment that the CT was performed or was there a time difference between RT-PCR test and CT?
Thank you for this important comment. Both were done at the time of ED arrival, first the RT-PCR immediately followed by the chest CT. We precise this point on line 76-77.
Results P3 line 32: It would be interesting to know when the second test was performed and when the 32 negative patients finally tested positive
The number and the date of RT-PCR performed during the stay were very heterogeneous depending on the admission in the ED t. Some patients did not have another nasopharyngeal swab while others have had several during the hospital stay, and those regardless of the results of the previous swabs. Unfortunately, these data were not usable. During this crisis situation, all physicians considered as COVID-19 a typical clinical and radiological presentation. We precised this point on results, line 122-123.
It is not clear to me what the clinical value of the findings in this study is. Both groups had a clear diagnosis of Covid-19 pneumonia and adequate treatment could be started and adapted to the clinical evolution. In addition not so many differences between the two groups were found. It is also well known as stated in the introduction that a RT-PCR test can be negative while CT shows clear signs of Covid-19 pneumonia.
At the beginning of the crisis, COVID-19 was unknown, and this clinical question was very important question as you can guess. In our organisation we choose to consider patient with negative RT-PCR and typical presentation as COVID-19. We think that the results of this study confirm our disease management and we agree with your remark.
Best regards
For the authors
M Oberlin

Reviewer 2 Report
This is an interesting study on a very important topic. In these challenging times, we need to have a deep understanding of the accuracy of diagnostic tests of COVID-19 and how a positive or negative result affects treatment and prognosis.
I believe one challenge of the study might be the evidence of a selection bias, which should be critically mentioned by the authors. Also, how the authors defined the definite presence of COVID-19 should be discussed, so that the results can be interpreted better by the audience.
Abstract:
Concise and well written
Intro:
Only minor comments:
Line 31: I believe a word is missing.
Line 33: I believe the sentence should be rephrased. It is very important to accurately diagnose COVID-19 as the numbers increase. But the increase in numbers is no challenge to accurate diagnosis.
M&M:
Although Chest CT has a high accuracy for identifying signs of COIVD-19, it cannot be used for the definite diagnosis of disease. I think it is reasonable to strongly suspect COVID-19 in patients with typical signs in CT, but ultimately rely on RT-PCR for the final diagnosis. However, in line 72-74 the authors state that a typical chest CT OR positive RT-PCR were used for the definite diagnosis of disease.
Although we are in the middle of a pandemic, a negative RT-PCR and atypical infiltrations in CT will more likely be another type of infection (which are still prevalent). Especially in areas with a lower prevalence of COVID-19, this should be kept in mind.
As the authors stated, their site “was one of the most 58 impacted by the pandemic in Europe”, so the majority of cases with initial negative RT-PCR and COVID-19 might still have been true COVID-19 cases, but this might very well be the results of a selection bias.
Minor comments:
Line 62: small spelling mistake
Line 107-110: This is an interesting approach. I am curious why the authors chose this method? I believe P values are often better understood by the research community. How would you define significance? (e.g. like in line 127).
Results:
Line 117: Following up on my comment above, could it be that those patients were co-isolated and some indeed hat another infection with atypical infiltration in CT and thus have become infected by other patients in the course of time?
Table 1:
“breaths per minute, mean (SD) 23.8 (5.7) 27.1 (9.1) -3.16 [-5.42;-0.87] <0.01*”
“≥24 breaths per minute 21 (50.0) 99 (68.8) -0.18 [-0.35;0.02] 0.01 *”
“Probability >90% or <10% that difference is larger than 0.”
Does that mean, that the probability, that the difference in breathing frequency is larger than 0 is smaller than 1%? This does not seem right. Could the authors please explain this in more detail?
Discussion:
Line 7-13: The accuracy of Chest CT has been criticized strongly and it is not recommended as screening. Initial studies have probably suffered from strong selection bias. Please refer to the following studies for more information:
Adams HJA, Kwee TC, Kwee RM. COVID-19 and chest CT: do not put the sensitivity value in the isolation room and look beyond the numbers. Radiology. 2020:201709.
Rubin GD, Ryerson CJ, Haramati LB, Sverzellati N, Kanne JP, Raoof S, et al. The Role of Chest Imaging in Patient Management during the COVID-19 Pandemic: A Multinational Consensus Statement from the Fleischner Society. Radiology. 2020:201365.
Radiology ACo. ACR recommendations for the use of chest radiography and computed tomography (CT) for suspected COVID-19 infection. ACR website Advocacy-and Economics/ACR-Position-Statements/Recommendations-for-Chest-Radiography-and-CTfor-Suspected-COVID19-Infection Updated March. 2020;22.
Hope MD, Raptis CA, Shah A, Hammer MM, Henry TS. A role for CT in COVID-19? What data really tell us so far. The Lancet. 2020;395(10231):1189-90.
Please also discuss the points mentioned in the comments of the M&M section.
Author Response
Dear reviewer 2,
Please find attached the answers about your questions and remarks. We would like to thank you about your important comments which improved our manuscript.
Intro:
Only minor comments:
Line 31: I believe a word is missing.
Exact, we complete.
Line 33: I believe the sentence should be rephrased. It is very important to accurately diagnose COVID-19 as the numbers increase. But the increase in numbers is no challenge to accurate diagnosis.
We agree, we correct the whole sentence.
M&M:
Although Chest CT has a high accuracy for identifying signs of COIVD-19, it cannot be used for the definite diagnosis of disease. I think it is reasonable to strongly suspect COVID-19 in patients with typical signs in CT, but ultimately rely on RT-PCR for the final diagnosis. However, in line 72-74 the authors state that a typical chest CT OR positive RT-PCR were used for the definite diagnosis of disease.
Although we are in the middle of a pandemic, a negative RT-PCR and atypical infiltrations in CT will more likely be another type of infection (which are still prevalent). Especially in areas with a lower prevalence of COVID-19, this should be kept in mind.
As the authors stated, their site “was one of the most 58 impacted by the pandemic in Europe”, so the majority of cases with initial negative RT-PCR and COVID-19 might still have been true COVID-19 cases, but this might very well be the results of a selection bias.
We precise in M&M lines 67-70 the definition of COVID-19 in our study. During this crisis, in our organisation we choose to consider patient with negative RT-PCR and typical presentation (clinical + imaging) as COVID-19 because of very high prevalence of disease. We discuss this important point in study limitations (discussion, lines 49-54).Please inform us if you need more precision.
We precise line 82-83 on under that only typical chest CT were analysed to reduce the risk of selection bias as you suggest.
Minor comments:
Line 62: small spelling mistake
We corrected this comment
Line 107-110: This is an interesting approach. I am curious why the authors chose this method? I believe P values are often better understood by the research community. How would you define significance? (e.g. like in line 127).
Please find behind some explanations from our statistician team about the Bayesian method. Please precise if you need more precisions in manuscript.
Why Bayes ?
Bayesian analyses are an appropriate alternative to the frequentist methods and have several advantages.
The concept of Bayesian statistics relies on the Bayes theorem which simply states that to improve knowledge about a paremeter, an a priori probability distribution must be given to the parameter (the chances that the parameter lies in a given interval), which once combined with the data observed during the experiment yields a posterior distribution that contains all the relevant information about the parameter. For instance, in a prognostic study, in most cases, the OR can be anticipated to be between 0.5 and 2. Based on the data observed in a prospective study, the OR is computed to be in the [1.2 – 1.9] interval and the probability that this OR is larger than 1 is almost 1. It is also easy to compute the probability that the OR is larger than 1.5 for example, which would give the probability that the effect is relevant as judged by an effect at the population level.
We thus chose to use Bayesian statistical method because of their ability to include prior information in order to accumulate evidence for or against an effect. Indeed, bayesian methods provide a formal way to introduce prior information (which is crucial for instance in the context of uncommon disease and small sample size but is always usefull).
The parameter estimates are issued with its “credibility interval” that has the correct and intuitive interpretation that is almost always wrongly given to the (frequentist) confidence interval: a 95% credibility interval is an interval where the true value of the parameter (OR for instance) lies with a probability of 0.95.
Results are presented by describing posterior distributions, and conclusions are formulated in terms of probability for the coefficient to be strictly positive. Indeed, Bayesian methods can compute quantities such as Pr(θ > 1) which cannot be produced by a frequentist analysis. If this probability is close to 1 (resp. close to 0), we will conclude that the probability of the tested hypothesis is very high (resp. very small). If this probability is close to 50%, the distribution is half negative, half positive, and we’ll conclude that the coefficient is not different from 0. Conclusions are then more informative than simple p-values.
Bayesian results are usually given with what would be termed « a unilateral view » in the frequentist language. Thus, if Pr(OR>1) >0.95 is considered as significant, it is not contradictory with a credibility interval that cover the value « 1 » since a 95% credibility interval leaves 2.5% probability below and 2.5% probability above this credibility interval. When the Pr(OR>1) is larger than 95% but smaller than 97.5%, the result is significant (unilateral point of view) but the OR credibility interval (bilateral point of view). Those are only two different views of a same object, i.e. the posterior OR distribution, which in Bayesian statistics can be cut any number of ways, as desired.
Bayesian analyses do not use nor compute the (‘frequentist’) p-value and thus the probability of exceeding 1 should not be confused with a p-value.
Exact results are given, even for small sample size, contrary to frequentist methods which usually rely on asymptotic results and approximation. Small sample estimations are often erroneous. For instance, Bayesian methods avoid mathematically “side” problems such as proportions that are close to 0 where the confidence interval can sometime be outside the [0 ; 100%] interval. The same problem arises when the proportion is close to 1.
The Bayesian methods give the answer expected by the physician, Pr(effect| data), i.e. the probability that the effect is large according to the experimental data, while the p-value gives the probability of the observed data given a null effect (or a specific non null effect), a single null hypothesis. To switch from the second to the first result, you must use the Bayes theorem.
The power of a frequentist test is the probability to reject the null hypothesis if the alternative is true but the power is usually confused with the probability that the alternative is true which is not the same. Once again, to get the right number here, the Bayes theorem is required. In the frequentist context, Pr(H1) is unknown before the test and remains unknown after the test. Bayesian methods do not need any assumption on power to estimate proportions since they are not based on a null hypothesis H0.
A large and recent literature has underlined the strength of Bayesian methods in clinical research. Among them, here is a sample of the most relevant.
- Wijeysundera N, Austin PC, Hux JE, Beattie WS, Laupacis A. Bayesian statistical inference enhances the interpretation of contemporary randomized controlled trials. Journal of Clinical Epidemiology 2009;62:13-21.
- Spiegelhalter DJ, Myles JP, Jones DR Abrams KR. An introduction to bayesian methods in health techonoly assessment. BMJ 1999;319:508-512.
- Spiegelhalter DJ, Abrams KR, Myles JP. Bayesian approaches to clinical trials and health-care evaluation 2004, John Wiley & Sons, Chichester.
- Adamina M, Tomlinso G, Guller U. Bayesian statistics in Oncology. A guide for the clinical investigator. Cancer 2009;115(23)5371-81. doi: 10.1002/cncr.24628.
- Freedman L. Bayesian statistical methods. BMJ. 1996;313(7057):569-70.
- Gelman A, Hill J. Data Analysis using regression and multilevel - Hierarchical models 2006, Cambridge university press, Cambridge.
- Ntzoufras I. Bayesian modelling using WinBUGS 2009, John Wiley & Sons, New York.
- Hoff. A first course in Bayesian Statistical Methods (Springer).
Several papers also tackled the comparison between Bayesian and frequentist methods in health data. See for example,
- Cohen, J. 1994. The earth is round (p<.05). American Psychologist 49:997-1003.
- Dunson D. Commentary: practical advantages of Bayesian analysis of epidemiologic data. American Journal of Epidemiology 2001, 153(12): 1222-6.).
Finally an increased number of published papers reports Bayesian results. See for a clear example:
- Morris R, Malin G, Quinlan-Jones E et al. Percutaneous vesicoamniotic shunting versus conservative management for fetal lower urinary tract obstruction (PLUTO): a randomised trial. The Lancet 2013, 382:1496-1506).
Results:
Line 117: Following up on my comment above, could it be that those patients were co-isolated and some indeed hat another infection with atypical infiltration in CT and thus have become infected by other patients in the course of time?
We precise in methods that only patients with typical lesions of COVID-19 on CT scan were included in our study.
Table 1:
“breaths per minute, mean (SD) 23.8 (5.7) 27.1 (9.1) -3.16 [-5.42;-0.87] <0.01*”
“≥24 breaths per minute 21 (50.0) 99 (68.8) -0.18 [-0.35;0.02] 0.01 *”
“Probability >90% or <10% that difference is larger than 0.”
Does that mean, that the probability, that the difference in breathing frequency is larger than 0 is smaller than 1%? This does not seem right. Could the authors please explain this in more detail?
The absence of difference is represented by a probability of 0.5. Indeed, a 50% probability that there is a difference greater than 0 is no better than chance.
Conversely, a probability of difference greater than 0, i.e. close to 0 or to 1, suggest an effect, respectively of a positive or a negative difference. Regarding the respiratory rate, the difference between the PCR- group and the PCR + group is negative. The probability of having a difference >0 is therefore <1%. In other words, the probability of having a negative difference is> 99%.
Discussion:
Line 7-13: The accuracy of Chest CT has been criticized strongly and it is not recommended as screening. Initial studies have probably suffered from strong selection bias. Please refer to the following studies for more information:
Adams HJA, Kwee TC, Kwee RM. COVID-19 and chest CT: do not put the sensitivity value in the isolation room and look beyond the numbers. Radiology. 2020:201709.
Rubin GD, Ryerson CJ, Haramati LB, Sverzellati N, Kanne JP, Raoof S, et al. The Role of Chest Imaging in Patient Management during the COVID-19 Pandemic: A Multinational Consensus Statement from the Fleischner Society. Radiology. 2020:201365.
Radiology ACo. ACR recommendations for the use of chest radiography and computed tomography (CT) for suspected COVID-19 infection. ACR website Advocacy-and Economics/ACR-Position-Statements/Recommendations-for-Chest-Radiography-and-CTfor-Suspected-COVID19-Infection Updated March. 2020;22.
Hope MD, Raptis CA, Shah A, Hammer MM, Henry TS. A role for CT in COVID-19? What data really tell us so far. The Lancet. 2020;395(10231):1189-90.
Please also discuss the points mentioned in the comments of the M&M section.
We agree partly with your comment and precise this point in limitations (Discussion).
We have also added this freshly published study in Radiology, aggregating a national French experience and advocating strongly the efficiency of chest CT as a screening tool in an epidemic context:
Herpe, G., Lederlin, M., Naudin, M., Ohana, M., Chaumoitre, K., Gregory, J., ... & Ludwig, M. (2020). Efficacy of Chest CT for COVID-19 Pneumonia in France. Radiology, 202568.
We integrate the references and hope message will be clear for readers.
Best regards
For the authors
M Oberlin

Round 2
Reviewer 2 Report
Dear authors,
thank you for this informative response letter and the revision of the manuscript. All my questions have been answered and I have no further comments. I approve this paper for publication in JCM.
best regards